# The importance of choosing appropriate methods for assessing wild food plant knowledge and use: A case study among the Baka in Cameroon

Sandrine Gallois[1]*, Thomas Heger[2], Amanda Georganna Henry[1], Tinde van Andel[2,3]

**1** Faculty of Archaeology, Leiden University, Leiden, The Netherlands, **2** Biosystematics Group, Wageningen University, Wageningen, The Netherlands, **3** Naturalis Biodiversity Center, Leiden, The Netherlands

\* galloissandrine.ethnoeco@gmail.com

## Abstract

In tropical rainforests, access to and availability of natural resources are vital for the dietary diversity and food security of forest-dwelling societies. In the Congo Basin, these are challenged by the increasing exploitation of forests for bushmeat, commercial hardwood, mining, and large-scale agriculture. In this context, a balanced approach is needed between the pressures from forest exploitation, non-timber forest product trade and the livelihood and dietary behavior of rural communities. While there is a general positive association between tree cover and dietary diversity, the complex biocultural interactions between tropical forest food resources and the communities they sustain are still understudied. This research focuses on the knowledge and use of wild food plants by the forest-dwelling Baka people in southeast Cameroon. By using two different sets of methods, namely *ex-situ* interviews and *in-situ* surveys, we collected ethnographic and ethnobotanical data in two Baka settlements and explored the diversity of wild edible plants known, the frequency of their consumption, and potential conflicts between local diet and commercial trade in forest resources. Within a single Baka population, we showed that the *in-situ* walk-in-the-woods method resulted in more detailed information on wild food plant knowledge and use frequency than the *ex-situ* methods of freelisting and dietary recalls. Our *in-situ* method yielded 91 wild edible species, much more than the *ex-situ* freelisting interviews (38 spp.) and dietary recalls (12 spp.). Our results suggest that studies that are based only on *ex-situ* interviews may underestimate the importance of wild food plants for local communities. We discuss the limitations and strengths of these different methods for investigating the diversity of wild food plant knowledge and uses. Our analysis shows that future studies on wild food plants would profit from a mixed approach that combines *in-situ* and *ex-situ* methods.

**Data Availability Statement:** All relevant data are within the manuscript and its Supporting Information files.

**Funding:** This research has received funding from the European Research Council under the European Union's Horizon 2020 research and innovation program; grant agreement number STG–677576 ("HARVEST") awarded to AGH. The botanical fieldwork was funded by Naturalis Biodiversity Center (https://www.naturalis.nl/en)to TVA, the TreubMaatschappij and the Alberta Mennega Stichting (https://www.alberta-mennega-stichting.nl) to TH. None of the funders has played a role in the research design, data collection, analysis, decision to publish or preparation of the manuscript.

**Competing interests:** The authors have declared that no competing interests exist.

## Introduction

In tropical rainforests, access to and availability of natural resources are vital for the dietary diversity and food security of forest-dwelling societies. The Congo Basin includes the largest tract of tropical rainforest in Africa. This highly biodiverse region is inhabited by more than 30 million people, including forest-dwelling communities that strongly depend on their natural environment for their survival [1]. A wide variety of wild edible plant species (WEP) has been reported for the Congo Basin, among Bantu-speaking farmers and especially among hunter-gatherers that infrequently practice agriculture [2–6]. Local people harvest these wild food plants mainly for subsistence use within the household, but a substantial number of species is locally traded, and a few find their way to global markets [7, 8]. The cash income earned with the trade in non-timber forest products (NTFPs), important for improving livelihoods, can empower local communities [9]. Wild edible plants also provide a safety net in times of agricultural shortage and are thus essential for food security [9]. However, forest-dwelling societies in the Congo Basin are challenged by a decrease in access to and availability of wild food resources due to the increasing exploitation of the forests for commercial hardwood, precious minerals, and space for oil palm and rubber plantations [10, 11]. As a result of large-scale agricultural encroachment, logging, mining and hunting activities, the ecosystems of this region are under severe pressure [10, 12]. A recent survey estimated that just 70% of the standing forests of the Congo Basin remain fully intact [11]. Pressures on local resources have led to a decrease in local people's access to the important wild plants and bushmeat, which contributes substantially to their diet and medicine [13, 14]. In Cameroon, for example, more than 60% of the top 23 timber species exported from the country also yield important NTFPs for local people. A balanced approach is needed between tropical hardwood exploitation and the livelihoods of rural communities, for which the development and implementation of sustainable forest management plans are essential [9].

Although the value of local ecological knowledge for informing conservation and environmental management is well established [15–17] and there is a general positive association between tree cover and dietary diversity [18], the complex biocultural interactions between tropical forest food resources and the communities they sustain are still understudied [10, 19]. According to Ngome et al. [20], in most poor communities in the Congo Basin, the wide variety of nutrient-rich, wild edible plants are underutilized, either due to ignorance, shame or misconceptions about these foods. Ethnobotanical studies have so far insufficiently provided evidence for the intake, quality and nutritional impacts of WEPs, their contribution to the overall diet and their role in ensuring food security [20]. Considering the increasing conflicts of interests regarding the use of wild resources, accurate assessments of local uses and dependency on wild plants and animals are necessary to evaluate the effects of commercial forest exploitation on both biodiversity and local livelihoods. The aim of this research was to explore the diversity of wild edible plants known and used by the forest-dwelling Baka people in southeast Cameroon. We also evaluated the frequency of their consumption, potential conflicts between local diet and commercial trade in forest resources, and the different field methods to assess of WEP knowledge and consumption.

The Baka are Ubangian-speaking forager-horticulturalists, formerly referred to as 'Pygmies' [21]. Until roughly 50 years ago, they were nomadic foragers who relied on hunting, fishing, gathering, and the exchange of NTFPs for agricultural crops with their neighbors, Bantu-speaking farmers [22]. Since the 1960s, the Baka have been facing several changes in their livelihood. Due to missionary activities and a government program of sedentarization, they have progressively left their forest camps and settled in villages along logging roads [23]. Nowadays, Baka livelihood is mostly based on the combination of foraging activities, agricultural work in

their own fields and wage labor on Bantu farms or for logging companies [22]. The settlement programs, which included displacement from protected areas and forced sedentarization, have had drastic effects on Baka livelihood and wellbeing. Increased alcohol abuse, the loss of traditional knowledge, and an increasing destabilization of their natural environment have led to the marginalization of the Baka and their increased dependence on other ethnic groups [24, 25]. To evaluate the Baka's reliance on their surrounding forests in the current situation, it is essential to assess not only their knowledge on wild food plants, but also their actual use and commercialization of these species.

Local knowledge and use of plants are assessed through different ethnobotanical methods, of which the interview is the most widely used. Interview methods vary considerably between studies and are deployed based on the research question addressed [26]. Freelisting is a frequently used interview method, in which informants are asked to list all items they know within a given domain [27, 28]. This technique reveals cultural salience and variations in individuals' topical knowledge [29], and results in a shortlist of highly valued plants [30, 31]. As freelisting enables the collection of data from a large number of informants in a limited amount of time [32], this method is frequently used as a starting point for studying traditional plant knowledge. Local names of plants listed during the freelisting interviews are identified afterwards by collecting voucher specimens for these names. The resulting dataset is then used to draw conclusions about plant knowledge by a certain group of people and/or the potential contribution of wild plants to their diet [30, 33]. Organizing field trips to collect herbarium specimens of species mentioned during freelisting exercises is often (inaccurately) called the 'walk-in-the-woods method' [34, 35]. However, this technique, first coined by Phillips and Gentry in 1993 [36], implies that participants are encouraged to actively lead field trips and point out all useful plants they know and/or use [26], instead of only helping the researcher search for specimens that can be linked to the local names mentioned during interviews. Data elicited from freelisting appear to be specific to the context in which they were collected (e.g., in the village), creating an unintended but significant bias in this type of ethnobotanical research [27, 32, 37]. Gathering the data during *ex-situ* interviews, for example in people's home, away from the ecological context in which people collect their plants, may result in lists of only the most salient plant species. Furthermore, the success of freelisting depends on the informants' correct understanding of the category or cultural domain (e.g., wild food plants) under discussion [28, 29, 38].

In societies that undergo rapid socio-economic changes, people become more integrated into the market economy, change their lifestyle and adopt cultivated or processed substitutes for wild plants in their diet [39]. This creates a gap between people's ethnobotanical knowledge and their actual use of plants [40, 41]. A discrepancy between the number of useful species known and those actually used indicates that elders who still know how plants were used in the past do not practice this any longer, and infrequently transfer their skills to the next generation [40]. Freelisting exercises often focus on people's knowledge [40], while recall surveys, developed by social anthropologists for understanding time allocation [42], lead informants to enumerate what they have done during a specific period of time. For instance, dietary recall surveys have been developed to estimate the proportion of different food items in people's diet [43]. Recently, dietary recalls were introduced in ethnobotanical studies to assess local uses of plants [44, 45], while income recall surveys have been used to assess the contribution of the sale of different forest products to local livelihoods [46]. Although many studies reported a high diversity of wild edible plant species worldwide [47, 48], research relying on dietary recalls has resulted in surprisingly low numbers of wild species actually being consumed [49, 50]. Dietary recalls carried out in the Democratic Republic of the Congo (DRC) also showed that wild plants did not contribute substantially to rural and urban women diets [51]. Likewise,

the dietary recalls that we recently held among the Baka reported only 15 wild edible species [52], which is much less than the list provided recently in the northern Congo area [53] and in stark contrast to the extensive wild plant knowledge documented almost three decades earlier by anthropologists among the same ethnic group [2, 3]. Like freelisting, dietary recalls are limited by the subject's memory [54] and may therefore underreport plant use.

Whether the limited amount of WEP listed by the Baka was a result of a general loss of traditional plant knowledge, a change in food preferences or the effect of the assessment methods, was essential to contextualize our findings. In this study, we explored how different ethnobotanical methods captured the diversity of wild edible plant knowledge and use in a Baka community. We aimed to answer the following questions:

1) Which wild edible plants (WEP) are known and used by the Baka?

2) What are the most frequently consumed WEP species among the Baka?

3) Do commercial logging and NTFP trade affect the availability of wild food plants?

4) How do the different methods for assessing WEP used in this study vary in their results?

## Materials and methods

### Study site

We collected data on wild food plants in two Baka settlements: Le Bosquet (3˚07'38"N13˚ 52'57"E) and Kungu (3˚02'40"N 14˚06'57"E), Haut Nyong division, southeastern Cameroon. The two settlements, which belong to the same population of Baka hunter-gatherers, are both located at least eight hours by car from the capital Yaoundé, of which four hours on unpaved logging roads (Fig 1). The accessibility of this area highly depends on the weather, as the road quickly deteriorates during the rainy season. The area is covered by a mixture of evergreen and moist semi-deciduous forest within altitudinal ranges of 300–600 meters [55]. In populated areas, the forest cover is largely removed in favor of settlements, cocoa plantations, logging activities and small-scale agriculture. This creates a mosaic of dense primary forest, selectively logged primary forest, secondary forest and agricultural fields, interspersed with trails. The climate of the region is tropical humid, with a major rainy season between late-August and late-

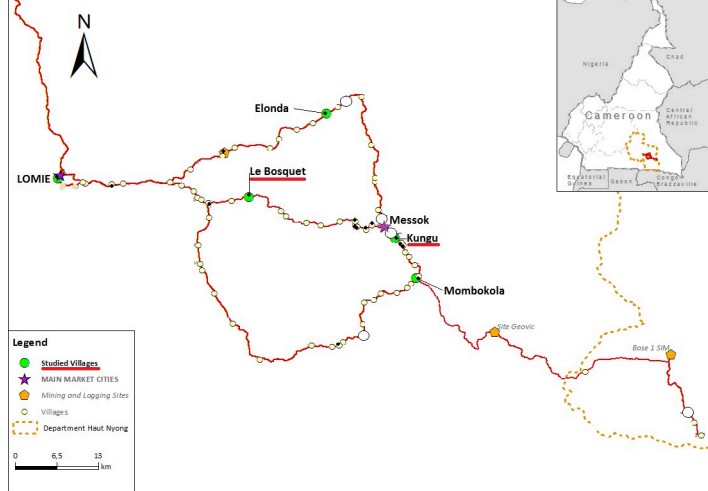

**Fig 1. Map of the study area.**

November and a major dry season between late-November and mid-March. The annual precipitation reaches about 1500 mm and the average temperature is 25°C [55].

Before data collection, Free Prior and Informed Consent was obtained from all participants. This study adheres to the Code of Ethics of the International Society of Ethnobiology [56], and received approval from the ethics commission of Leipzig University (196-16/ek) and the Ethical Committee from the Ministry of Health of Cameroon (n°2018/06/1049/CE/CNERSH/SP). For our inventory of wild food plant knowledge, their use and commercialization, we used a combination of four different datasets, obtained from *ex-situ* interviews (freelisting, dietary recalls and income recalls) and *in-situ* ethnobotanical forest surveys ('walk-in-the-woods' method). We collected data in the two Baka settlements in 2018 and 2019, during three different fieldwork periods, to cover variations in wild fruit availability (Table 1).

## Data collection: *Ex-situ* interviews

All data were collected among Baka individuals of 18 years and older. We included all individuals who were present in the settlement and willing to participate. While we aimed to get a gender- and age-balanced sample, we did not exclude any potential participants. All *ex-situ* interviews were held in the French and Baka language by the first author with the assistance of a local research assistant-interpreter. During the 55 freelisting exercises, we asked 24 men and 31 women in the two Baka villages (29 in Le Bosquet, 26 in Kungu, belonging to 51 different households) to report all WEPs they knew [52]. Because no specific term exists in Baka language that refers to 'wild edible plant', we asked them: "Which food from the forest, excluding game, honey and mushrooms, do you know?" (S1 File). To gather data on the actual consumption of wild food plants and their importance in Baka diet, we conducted a dietary recall protocol that we adapted from the FAO Guidelines for Assessing Dietary Diversity [52, 57]. Participants were asked to list all items they had consumed within the previous 24 hours, and to mention the origin of each food item (from the wild, from agricultural fields or bought at the market). A total of 143 dietary recall interviews were conducted in the two settlements among 83 individuals: 35 men and 48 women (38 in Le Bosquet and 45 in Kungu), belonging to 66 different households; 42 individuals were interviewed once, 22 twice and 11 three times (S1 File). A detailed account on the outcomes of the freelisting and dietary recalls with regard to cultivated and shop-bought food, preferences and perceptions of wild vs. cultivated food was published earlier [52].

**Table 1. Datasets, fieldwork periods and the different methods used to assess wild food plant knowledge, consumption and commercialization among the two Baka settlements.**

| Fieldwork period | Feb.-March 2018 | Oct.-Nov. 2018 | April-May 2019 | |
|---|---|---|---|---|
| Datasets and methods (number of participants = n) | major dry season | major rainy season | minor dry season | Data published earlier by authors |
| 1. Freelisting(n = 55) | x | | | [52] |
| *ex-situ* | | | | |
| 2. Dietary recalls (n = 83) | x | x | x | [52] |
| *ex-situ* | | | | |
| 3. Income recalls (n = 73) | x | x | | [52] |
| *ex-situ* | | | | |
| 4. Walk-in-the-woods survey | | | x | [71]* |
| 14 days (n = 20) *in-situ* | | | | |

* Only data regarding bush mangoes (Irvingiaceae).

To collect data on wild food plants that were commercialized (either traded as timber or as NTFP), we conducted a 14-day recall survey on the income received through sale, asking 73 participants to list all the items they had sold during this time period [58]. A total of 114 interviews were conducted with43 women and 30 men (34 individuals in Le Bosquet and 39 in Kungu): 32 individuals were interviewed once and 41 twice (Table 1). Parts of the *ex-situ* interviews were repeated in other periods of the year to verify whether the consumption and/or commercialization of WEP changed according to the season. None of the three *ex-situ* interview methods were based on a predefined set of species: we did not use pictures of wild food plants, fresh samples or other visual stimuli, and neither gave prompts after informants had listed a certain number of plants.

## Data collection: *In-situ* walk-in-the-woods survey

From the local (Baka) names mentioned during the three *ex-situ* interview methods, we constructed a preliminary database of 'folk taxa' of wild plants consumed by the Baka, with tentative scientific names retrieved from literature on Central African wild food plants [2, 3, 6, 59] and earlier ethnographic and ethnoecological data collected between 2012 and 2014 by the first author during ca.18 months of fieldwork in the Baka settlements [22]. During our last fieldwork period (2019), we carried out a walk-in-the-woods survey to obtain botanical specimens to match the Baka names of wild food taxa in our preliminary database, but also to verify whether the Baka knew, consumed or traded more plant species than they mentioned previously during the *ex-situ* interview sessions. Therefore, we asked the Baka to suggest several people of different ages and gender who were knowledgeable on wild edible plants and would agree to join us on our collection trips. We worked with one to four informants on each collection day. In total, we worked with 20 individuals (10 women and 10 men), aged between 29 and 80 years, of which nine had participated previously in the *ex-situ* interviews (two in the dietary and the income recalls; two only in the free listing; five in all three methods). During 14 collection days into the area surrounding Le Bosquet and Kungu, we searched for plants that matched our preliminary list of folk taxa, but also asked our informants to point out any other edible species they saw (S1 File). When such a plant was encountered, we collected herbarium material using standard botanical methods [27]. Fieldtrips lasted from 7:00 am to 15:00 pm, but the distance covered differed per day, as it depended on the number of plants collected. Many more species were collected in the first few days close to the village, while several hours were spent walking to collect rare edible plants on the last days. We paid particular attention to clarify information on Baka folk taxa that consisted of more than one botanical species. For most specimens we collected, we asked our informants for 1) the local name (in Baka, French and/or Nzimé, the Bantu language in this area, if known); 2) plant part(s) used; 3) preparation and application methods; 4) when they had last consumed the plant; 5) whether a part of the plant was sold; 6) in the case of trees, whether the wood was commercially logged. We did not limit our ethnobotanical data to plant uses that were shared among our informants, but were keen to record differences in food plant knowledge, consumption frequency and appreciation of wild edibles among informants.

Duplicates of voucher specimens were deposited at the National Herbarium of Cameroon (YA) and at the herbarium of Naturalis Biodiversity Center (L). A third voucher was kept at the study site to verify local names and uses with Baka villagers during group discussions. Plant identification took place at Naturalis, using Central African herbarium specimens and literature, such as the Flora of West Tropical Africa [60], the Flore du Cameroun [61], and monographs on the tropical African flora [62–64]. This literature was also used to verify the vegetation types in which these wild food plants occurred naturally. For species that were

difficult to identify, we consulted botanical experts at Naturalis and abroad. Scientific names were updated using the online portal of Plants of the World Online [65].

## Data analysis

We first analyzed the overall diversity of the WEPs reported and collected through the different methods. To assess the general characteristics of wild species consumed by the Baka, information on life form, part used, habitat and commercial timber was categorized in a spreadsheet, after which the distribution of these traits could be quantified. We used the Smith index [66] to assess the saliency of the plant species reported during the free listing interviews. To analyze the actual consumption of wild food plants reported during the dietary recalls, we calculated the citation frequency of the WEPs mentioned as eaten during the previous 24 hours. To analyze consumption data collected during the walk-in-the-woods survey, we categorized the respondents' answers in the following categories: 1) consumed today/yesterday; 2) 3–7 days ago, 3) 2–4 weeks ago; 4) 1–12 months ago; 5) 1–2 years ago; 6) > 2 years ago; and 7) never [52]. We visualized the ranking of the most recently consumed species according to the walk-in-the-woods methods in a bar chart (Fig 2). We also calculated the most frequently sold WEPs from the overall list of plants reported as commercialized in the income survey.

In order to assess whether the full potential of the methods had been used (dietary recalls, freelisting and walk-in-the-woods), species accumulation curves [67] were produced by calculating the cumulative number of species that were reported after interviewing a certain number of informants (freelisting and dietary recalls) and after a certain amount of collection days (walk-in-the-woods method). Contrary to usual practice, data were not randomized before producing the curves, as several relevant features of the data would have been lost.

Finally, in order to analyze conflicts between commercial timber harvesting and the availability of wild food plants for the Baka, we traced evidence of wild fruit trees felled by timber companies by counting the number of logged tree trunks along the forest trails and on logging

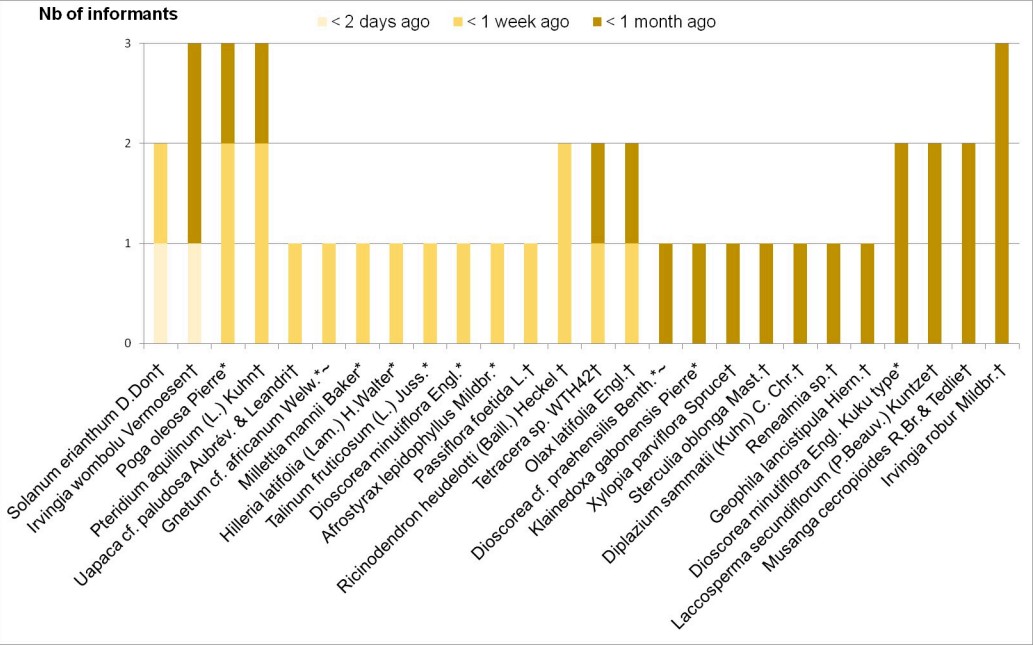

**Fig 2. Wild edible species consumed by the 20 informants within the past month.** *Species also reported during freelisting, ~Species also reported during dietary recalls, †Species not reported during either freelisting or dietary recalls.

trucks passing through the village during 14 days. We cross-referenced wild fruit trees species observed as felled logs and/or said to be cut for commercial timber with the CITES appendices [68] and the IUCN Red List [69] to assess their current conservation status.

## Results

### Diversity of wild edible plants

A total of 94 folk taxa of WEP were reported during the three different *ex-situ* interview methods and the *in-situ* walk-in-the-woods trips, corresponding to ca. 91 species. The exact number of species is unclear, as eight vouchers could only be identified to genus level and for several West and Central African *Dioscorea* species (wild yams), the taxonomic species delimitation is not clear [70]. Moreover, the Baka recognize different forms within individual yam species and thus some local names refer to the same botanical taxon. In the case of *D. minutiflora*, the Baka distinguish three distinct types: 'njàkàkà', 'báloko' and 'kuku', all with different leaf and tuber morphology. All local and scientific names of each wild edible species, used parts, preparation methods, and the method(s) through which they were recorded are listed in S1 Table. The 91 wild edible plant species belonged to 43 different plant families, of which the best represented were Dioscoreaceae (ca. 9 species of wild yams), Irvingiaceae (8 spp.), Anacardiaceae (5 spp., including 4 species of *Trichoscypha* fruits) and Zingiberaceae (5 spp. of *Aframomum*). Most wild food plants were trees (46%), followed by climbers (27%) including woody lianas and non-woody vines, herbs (19%), shrubs (7%) and ferns (1%). More than half (52.1%) of the wild edible plant species collected by the Baka naturally occurred in primary forest, 31.9% in secondary forests and the rest (16%) in open vegetation. We encountered very little primary forest that was untouched by loggers: the only patch of forest that did not show signs of commercial timber harvesting was dominated by *Gilberiodendron dewevrei*, located at ca. two hours walking distance from Le Bosquet. The selectively logged primary forest, however, contained the majority of the fruit and seed producing primary trees and lianas sought after by the Baka.

Of the 94 folk taxa of wild food plants, 38 were reported by the Baka during the free listing exercises. Initially, 51 local names were mentioned during these interviews, but 13 of those were later excluded because they were either synonyms of Baka plant names that had already been mentioned (three names) or they referred to wild mushrooms (two names), types of honey (six names) or cultivated plants (two names). The most salient WEPs (the most frequently listed first) emerging from the freelisting (Smith index> 0.1) were *Dioscorea burkilliana*, *D. praehensilis*, *D*. cf. *praehensilis* ('ba'), *Irvingia gabonensis*, *Baillonella toxisperma*, and *Panda oleosa*. Of the 83 participants of the dietary recalls, 69 reported having eaten wild plants, and mentioned a total of 12 different species. Two species that emerged from the dietary recalls (*Amaranthus dubius* nd *Raphia* sp.) were not mentioned during the freelisting.

During the dietary recalls, 12 species of wild edible plants emerged, of which the most frequently eaten was *Gnetum* cf. *africanum* (mentioned in 50% of the interviews). The identification to species level of *Gnetum* was difficult due to the absence of flowering material from male and female individuals during the time of our botanical survey. After *Gnetum* leaves, the second most often consumed were bush mango kernels (Irvingiaceae spp.; 22% of the interviews), *Raphia* palm wine (15%) and several species of wild yams (*Dioscorea* spp.; 15%). Detailed data on the contribution of WEPs to Baka diet compared to cultivated crops and store-bought food were published earlier [52].

During our walk-in-the-woods survey, much more WEP species were reported by the 20 informants as recently eaten. Of the 82 WEP species for which we collected information on last consumption during the walk-in-the-woods survey, 26 were reported as consumed within

the last month by at least one informant (Fig 2). The most recently consumed was *Solanum erianthum*: the bitter fruits of this shrub were boiled with wild garlic bark (*Afrostyrax lepidophyllus*) and (cultivated) pepper (*Capsicum frutescens*) and taken as a hot drink to wake up in the morning. Moreover, 36 species were eaten within the last 12 months, 11 species between one and two years ago, eight species more than two years ago and one species was never eaten by any of our 20 informants (Fig 2). Of the 26 species consumed within the last month, 23 would not have surfaced from the dietary recalls only, and 16 would have been missed if only the freelisting and dietary recalls would have been carried out. These 16 edible species included two ferns (*Pteridium aquilinum* and *Diplazium sammatii*), three spices (*Xylopia parviflora*, *Olax latifolia* and *Ricinodendron heudelotti*), four fruits (*Passiflora foetida*, *Solanum erianthum*, *Musanga cecropioides* and *Uapaca* cf. *paludosa*), the inner stem of *Laccosperma secundiflorum*, three seeds (*Sterculia oblonga*, *Irvingia robur* and *I. wombulu*), one root (*Renealmia* sp. WTH64), the potable water from the stem of *Tetracera* sp. (WTH42) and the leaves of *Geophila lancistipula*, eaten as a luck charm.

## Plant parts used and their preparation methods

Different parts of the wild food plants were eaten: most of them were fruits (37% of the species) and seeds (27%), followed by leaves (19%), tubers (12%), bark (5%) and exudate (1%). For 12 species, more than one part was consumed, such as fruits and seeds of *Aframomum* spp., *Irvingia gabonensis*, *Diospyros* cf. *crassiflora*, and *Baillonella toxisperma*; fruits and stem sap of *Musanga cecropioides* and *Chrysophyllum albidum*; fruits and leaves of *Piper umbellatum* and *Olax latifolia*; fruits and roots of *Dioscoreophyllum cumminsii*, and bark and seeds of *Afrostyrax lepidophyllus*. Most WEPs were integrated to cooked meals: only 36% of the 108 edible plant parts were eaten raw, but preparation methods depended on the part(s) consumed (Table 2).

The daily meal of the Baka was generally composed by two main dishes: a starchy basis and a stew with a fatty component, leaves, meat or fish. The carbohydrates provided by wild plants came predominantly from wild yams (*Dioscorea* spp.). The collection of these tubers required expert knowledge, as the Baka first verified whether the plant held tubers that were ready to be consumed, which in the case of *D. semperflorens* was done by examining the color of its leaves. The digging of the tubers required certain tools, like a wooden auger to remove the soil and extract the tuber without damaging it, a tool used specifically for both folk taxa of *D. praehensilis*. Once dug out, the tubers were washed and boiled. Some wild yams were only consumed occasionally as an emergency food, like 'boli' (*Dioscorea* sp. WTH63), of which the name was translated as 'adulterer', referring to the shame felt when eating this tuber. Other occasionally consumed wild tubers provided the Baka with carbohydrates, like those of *Palisota* cf. *barteri* and *Dioscoreophyllum cumminsii*.

To accompany this starchy basis, the Baka prepared a stew in which leafy vegetables and condiments were combined with a fatty basis, mostly obtained from forest seeds. Fruits of *Irvingia* spp., *Baillonella toxisperma* and *Panda oleosa* were gathered from the ground and the

**Table 2. Plant parts eaten and their preparation methods.**

| Part used | Fruits | Leaves | Tubers | Seeds | Sap | Bark | Stem | Total |
|---|---|---|---|---|---|---|---|---|
| Number of species | 38 | 20 | 12 | 31 | 4 | 2 | 1 | 108* |
| Eaten raw | 31 | 1 | 0 | 3 | 4 | 0 | 0 | 39 |
| Eaten cooked | 7 | 19 | 12 | 25 | 0 | 2 | 1 | 66 |
| Eaten raw and cooked | 0 | 0 | 0 | 3 | 0 | 0 | 0 | 3 |

*Total number exceeds the 91 species, as some wild food plants yielded more than one edible part.

kernels were taken out. Most seeds were prepared in a similar way: the kernels were dried and/ or roasted, and then pounded and/or ground to a powder that was mixed into the stew. Seeds could also be conserved for some time in the form of a pressed, oily cake. Leaves were either cut into thin slices (e.g., *Gnetum* cf. *africanum*) or pounded in a mortar (e.g., *Hilleria latifolia*) and added to the stew. Condiments were grated (*Afrostyrax lepidophyllus* bark), dried (*Aframomum* spp. seeds), or roasted and crushed (seeds of *Ricinodendron heudelotii* and *Monodora myristica*) before being added to the dish.

Some wild food plants were eaten only on specific occasions, mostly for ritual purposes. To prepare men before hunting expeditions and increase their success, women prepared the leaves of *Anchomanes difformis*, which were thought to make the hunter fearless for elephants, or the rhizome of *Renealmia* sp. (WTH64), which was thought to bring good luck to the hunter. The leaves of *Costus englerianus* or *Adenia mannii* were prepared to strengthening boys after their circumcision.

## Commercialization of wild edible plants

During the recall exercise on general income received from sale over the 14 previous days, the Baka reported six different taxa as being sold. The most frequently commercialized NTFP was *Gnetum* cf. *africanum* (26% of the 114 interviews), far more than any of the other wild food plants, which were mentioned as sold in less than 5% of the interviews (seeds and oil of *Irvingia gabonensis*, *Baillonella toxisperma*, *Pentaclethra macrophylla*, bark of *Afrostyrax lepidophyllus* and the fruits of *Aframomum* spp., indicated under the general name of 'tondo'). During the walk-in-the-woods surveys, however, the Baka pointed out 24 different WEP species that they sold to middlemen (including the six previously mentioned), mostly in the form of fruits, seeds, or the oil from seeds. Of the 24 commercial species that appeared during the forest surveys, 13 were commonly sold on the international NTFP market (Table 3). When we collected specimens during the walk-in-the-woods surveys, we discovered that the Baka name 'tondo' referred to four different species of *Aframomum*: *A. sceptrum*, *A. daniellii*, *A. subsericum* and *A.* cf. *longipetiolatum*. Only *A. cereum* was not sold as spice to middlemen, but just consumed within the household. Of the 24 commercial species traded by the Baka, four appeared in the IUCN Red List as vulnerable and two as near threatened (Table 3).

During our forest walks, we identified six WEP of which the wood was observed as logged or said to be logged by the Baka (Table 4). Three of these species were considered as vulnerable by the IUCN, but none appeared on the CITES Appendices I or II. The moabi tree (*Baillonella toxisperma*), highly valued by the Baka for their fresh fruits and seed oil, was the most sought after by the logging companies operating in Baka territory. Our informants mentioned that only trees exceeding one meter in diameter were felled, so several smaller individuals were still present. However, the extraction of large moabi trees by timber companies strongly affected the amount of fruits and seeds that remained available for the Baka's subsistence and cash income. Another species that we observed as felled trunks was *Entandrophragma cylindricum*, which itself was inedible, but commonly hosted edible caterpillars, an important food for the Baka. During the forest walks, we also observed several (smaller) trees cut down by the Baka themselves, mostly to obtain fresh leaves of *Gnetum* cf. *africanum* lianas, to harvest honey, and once to collect the bitter bark of *Garcinia kola*, which was added to *Raphia* palm wine as a flavoring agent.

## Discussion

### High diversity in wild food plants

A major insight provided by this study is the high diversity of WEPs known and eaten by the Baka (94 folk taxa, 91 species). Such results are comparable to earlier ethnographic studies

**Table 3. Wild food plant products sold by the Baka and their conservation status.** Data were retrieved through different methods.

| Species | Plant parts | Walk-in-the-woods | Income recalls | (Inter-) national trade | IUCN status |
|---|---|---|---|---|---|
| *Afrostyrax lepidophyllus* | bark | y | y | [8] | Vulnerable |
| *Irvingia gabonensis* | fruits, seeds | y | y | [8] | Near threatened |
| *Panda oleosa* | oil from seeds | y | | [7] | Least concern |
| *Gnetum* cf. *africanum* | leaves | y | y | [8] | Near threatened |
| *Dioscorea* cf. *praehensilis* | tuber | y | | No data | Least concern |
| *Baillonella toxisperma* | fruits, oil from seeds | y | y | Oil [7] | Vulnerable |
| *Pentaclethra macrophylla* | oil from seeds | y | y | [7] | Least concern |
| *Garcinia kola* | bark | y | | [8] | Vulnerable |
| *Piper guineense* | fruits | y | | [71] | Least concern |
| *Parinari excelsa* | firewood | y | | No data | Least concern |
| *Xylopia parviflora* | fruits | y | | [8] | Not evaluated |
| *Irvingia robur* | seeds | y | | No data | Least concern |
| *Ricinodendron heudelotii* | seeds | y | | [8] | Vulnerable |
| *Cola acuminata* | seeds | y | | [8] | Least concern |
| *Tetrapleura tetraptera* | fruits | y | | [8] | Least concern |
| *Laccosperma secundiflorum* | stem (craft material) | y | | No data | Least concern |
| *Aframomum* cf. *longipetiolatum* | fruits | y | y* | [8] | Not evaluated |
| *Aframomum subsericum* | fruits | y | y* | No data | Not evaluated |
| *Aframomum daniellii* | fruits | y | y* | No data | Least concern |
| *Aframomum sceptrum* | fruits | y | y* | [8] | Not evaluated |
| *Trichoscypha* sp. WTH25 | fruits | y | | [71] | - |
| *Monodora myristica* | fruits | y | | [71] | Least concern |
| *Piper umbellatum* | fruits | y | | No data | Not evaluated |
| *Solanum erianthum* | fruits | y | | No data | Not evaluated |

\* all mentioned under the Baka folk taxon 'tondo'; y = yes, reported in the respective interviews.

carried out among small forest dwelling societies in the Congo Basin [2]. Even though the Baka are facing socio-ecological changes and potential dietary transition [45], our results show that they still know and consume a large diversity of wild edible plants. Apart from the fact that the Baka no longer processed and consumed the oil of *Pentaclethra macrophylla*, but rather sold the seeds to middlemen who processed them elsewhere, we found little evidence for loss of knowledge or disappearing plant uses [52]. Most wild food plants occurred in (selectively logged) primary forests, and consisted predominantly of fruits and seeds of canopy trees

**Table 4. Commercial hardwood tree species producing edible fruits and/or seeds consumed by the Baka, trade names and current conservation status.**

| Scientific name | Baka name | Commercial trade name | Nr. logs observed* | IUCN status |
|---|---|---|---|---|
| *Baillonella toxisperma* | Mabe | Moabi | 9 | Vulnerable |
| *Chrysophyllum lacourtianum* | Bambu | Longhi, Abam | | Not evaluated |
| *Diospyros* cf. *crassiflora* | Lembe | (Gabon) Ebony | 2 | Vulnerable |
| *Trichoscypha* cf. *abut* | Agbo | - | | Least concern |
| *Desbordesia insignis* | Ntuo | Alep | | Not evaluated |
| *Sterculia oblonga* | Egboyo | Eyong | | Vulnerable |
| *Afzelia* sp. | Tanda | Doussier | 3 | No data |

\* Observed during 14 days by the authors. When no logs were observed, the species were mentioned by the Baka as being logged by timber companies.

and lianas, underlying the importance of this ecosystem for Baka diet and culture. For their daily fat intake, the Baka depended heavily on the large fruits and seeds of the Irvingiaceae family, *Panda oleosa* and *Poga oleosa*. These species are typical to the Congo Basin forests, in which complex food and dispersal interactions take place between big terrestrial mammals (such as elephants) and the local flora [62, 72, 76]. Many of the oily seed-producing trees in Central Africa are ecological keystone species that are crucial for the survival of local wildlife [72], on which forest-dwelling groups such as the Baka rely on for meat [13].

The diversity of the wild plants eaten by the Baka covers a wide variety of plants parts, highlighting the role played by WEP in providing important nutrients that may not be found in planted crops and store-bought foods. This diversity is also targeted by traders in timber and non-timber forest products: at least 30 WEP species are either sold to middlemen and enter the (inter-) national NTFP market or are extracted by loggers, which sometimes causes conflicts between the Baka and commercial parties entering their territory. The various direct and indirect effects of logging and trade in NTFPs in this region do not only affect human food resources, but have an impact on the entire ecosystem.

## Different methods yield different results

Although our research was performed among a relatively small human population, our results show that different methods resulted in substantial differences in the collected data. The *ex-situ* interviews did not capture the full diversity of WEPs known, used and sold by the Baka. As shown in Fig 3, the species accumulation curve of the freelisting method flattened after interviewing 55 individuals. Typically, 14 of the 55 respondents did not report any WEP, which resulted in several flat sections in the curve.

The species accumulation curve for the dietary recall method flattened after interviewing 83 people (Fig 3). Between respondents 46 and 83, only three new species were mentioned, which suggests that interviewing more respondents would not have led to much more wild edible species being mentioned. Therefore, both freelisting and dietary recalls appeared to have captured most of the WEP diversity that was possible by these methods. In contrast, the species accumulation curve for the walk-in-the-woods method flattened somewhat after 11 days, but not completely (Fig 4). Our Baka informants indeed mentioned that there were additional rare species that could only be found after walking for hours in the forest. We know that at least

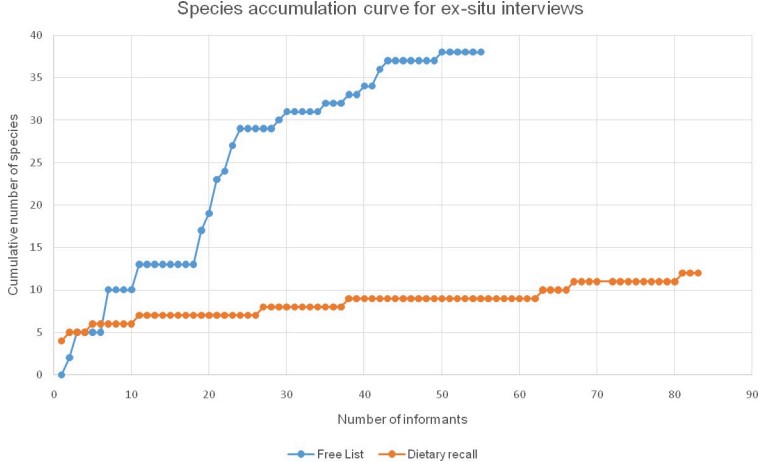

**Fig 3. Species accumulation curves of WEPs mentioned during the 55 freelisting and 83 dietary recall interviews in Le Bosquet and Kungu, southeast Cameroon, 2018.**

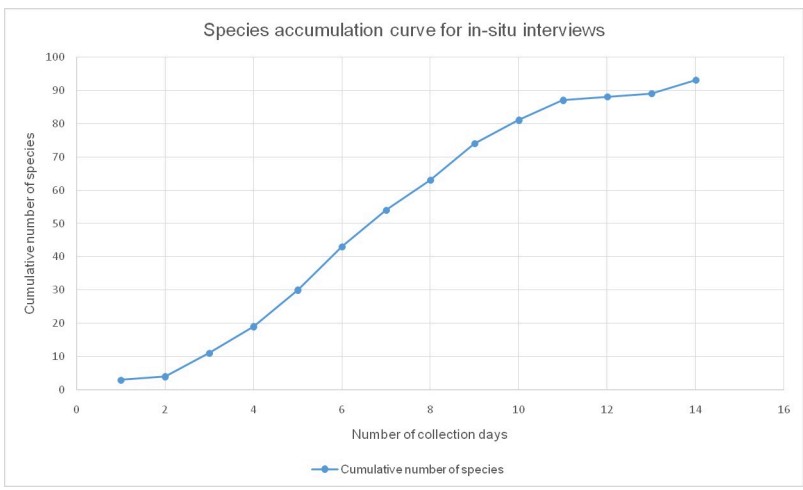

**Fig 4. Species accumulation curve of wild edible plants mentioned during 14 days of walking in the forest with 20 informants around Le Bosquet and Kungu, southeast Cameroon, 2019.**

four other edible species could have been found if we had more time to venture further into the forest. From their Baka names and the literature [2, 59] we assume that these were the African mammee apple (*Mammea africana*) with large edible fruits, a species of *Afzelia* of which the red arils around the seeds were eaten, a species of wild *Raphia* palm of which the sap was fermented into palm wine, and the African walnut tree (*Coula edulis*) that produced highly valued nuts.

Regarding the overall WEP diversity recorded through the different methods, we see that the botanical diversity reported would have been considerably less if we would only have conducted the *ex-situ* interviews. The freelisting (38 WEP species) and dietary recalls (12 spp.) resulted in just a fraction of the 91 species found during the walk-in-the-woods survey. Such low number of species reported may be partly due to the fact that not every participant understood the concept of 'wild edible plant'. Wild food plants play an important role in Baka livelihood [2, 3] and knowledge related to edible plants is acquired early during childhood [73]. Therefore, it seems unlikely that those Baka adult informants who did not report any WEP during the freelisting did not know any. They probably did not understand the domain, or the domain was too ubiquitous, a limitation that has been also found in other settings [74].

During the walk-in-the-woods method, the researcher can directly exclude items pointed out by informants that fall outside the domain 'wild edible plant', such as fungi, animal products and cultivated plants, although the latter category can be challenging due to the presence of wild species under various degrees and types of human management and domestication [48]. Local names listed as 'wild edible plant' during *ex-situ* interviews may later, during botanical collection trips, appear to refer to domesticated crops, like the cultivated American crop species tania (*Xanthosoma sagittifolium*) and chili pepper (*Capsicum frutescens*), which were listed as wild edibles during freelisting interviews in the DRC [34]. This was also the case in a recent study on WEP consumed by Baka south of the Dja Reserve [75], in which the domesticated crops okra (*Abelmoschus esculentus*), sorrel (*Hibiscus sabdariffa*) and chili pepper were listed as wild edible plants, as well as an unidentified yam with the Baka name mboto (*Dioscorea* sp.10), which was also collected by us and identified as the domesticated species *Dioscorea dumetorum*.

Our botanical inventory also revealed that a single folk taxon mentioned during interviews can actually include several species. The local Baka name 'tondo' referred to three different

species of *Aframomum*, the term 'bokoko' to two species of *Klainedoxa*, and 'payo' to three different species of *Irvingia* [76]. On the other hand, for the single taxon *Dioscorea minutiflora*, the Baka distinguished in three distinct types. The huge edible plant diversity collected in the forest survey was facilitated by our previously established list of folk taxa. If this preliminary work would not have been conducted, we would have needed more days of fieldwork and more informants to capture the full diversity of WEP consumed by the Baka.

While only six commercial wild food plants were recorded through the income recall surveys, the walk-in-the-woods method revealed 24 traded species, of which several had never been reported previously as sold in Cameroon. Moreover, this *in-situ* method allowed us to observe traces of logged WEP species and unsustainable ways of harvesting NTFPs. Only ten of the 26 species mentioned as recently consumed during the walk-in-the-woods method were reported during the freelisting, while just three of them also emerged through the dietary recalls, although the number of people interviewed during the last two methods was substantially higher. In other words, 23 recently consumed species would not have been identified with dietary recalls only, and 16 would have been missed if only the freelisting and dietary recalls would have been performed. We speculate that these were plants that were easily forgotten (e.g., spices, condiments, small fruits), species that people feel ashamed of eating (weeds), or items that were previously missed during the freelistings due to misinterpretation of the term 'wild edible plant' (e.g., drinking water from lianas, edible latex, ritual food plants). If we had only used the *ex-situ* interview data, we would have missed four wild fruit trees that were extracted by timber companies. Only two of these WEP-producing commercial hardwoods were mentioned in the freelisting interviews (*Baillonella toxisperma* and *Chrysophyllum lacourtianum*), while only one was recorded through the dietary recalls (*B. toxisperma*).

The advantage of assessing plant knowledge within the ecological context is that many species are encountered that do not pop-up quickly in people's minds during a (shorter) interview outside the forest. When walking through the natural environment where edible plants occur, it is easier to remember them because of the amounts of visual stimuli to this knowledge at that moment [77]. The walk-in-the-woods method, however, is laborious to perform and requires additional botanical collection, as more rare species will be encountered that are hard to identify, for which the help of taxonomic specialists and support from herbaria is needed. The number of informants that can be taken into the field is also limited and thus forest surveys are not suitable to draw conclusions on the general patterns of foraging, plant knowledge and uses.

On the other hand, *ex-situ* interview methods (including freelisting and dietary recalls) are known to assess the most salient useful plants among a large group of people in a relatively short time, as these techniques are limited by their spatiotemporal context [32, 37]. Freelisting yields much more information than just plant names and saliency, but also shows the items within a specific (local) domain and consensus in a community [74]. Conducted among a large number of people and in different seasons, dietary and income recall surveys provide a general overview of plants that contribute substantially to people's daily meals and subsistence strategies. Our results indicate, however, that these short *ex-situ* interviews missed out a substantial part of the WEP diversity known, consumed and traded by the Baka. Our insights show the importance of not relying on either *ex-situ* or *in-situ* methods, but rather combining both approaches to maximize the outputs assessing ethnobotanical practices, which was earlier highlighted by Quinlan [74].

Since they can lead to an underestimation of wild edible plants known, consumed and commercially exploited, the implications of studies based solely on *ex-situ* interviews can be grave. Results of such studies may not be representative for the situation on the ground, as trade in NTFPs or conflicts between wild fruit collection and logging of fruit-producing trees may

remain invisible. An assessment of Baka knowledge on wild edible plants relying on *ex-situ* methods only might have led to the conclusion that the Baka have lost much of their knowledge, or do not use the forest so intensively as their ancestors did. Moreover, the calculations of the contribution of wild plants to local diet and nutrition would have been inaccurate. Several studies based on dietary recalls have concluded that WEP do not play an important role in local diets. Termote et al. [51:8] said they were "confident to provide a fair representation of the dietary contribution of WEP on a population level in our sample", even though their botanical collection was limited to finding specimens to match the local names mentioned during their dietary recalls and freelisting interviews. In Brazil, scholars stated after their free-listing and dietary recall surveys that "the low consumption of wild species [. . ..] is notable, which suggests that, in practice, these foods contribute little to contemporary dietary enrichment" [50:337]. Such data could be misused by policy makers, who may conclude that rural communities do not need the forest that much as previously thought. Considering the importance of wild plants for food security and for providing nutrients that are not present in other foods [78], and the fact that children are major consumers of wild fruits but hardly recruited as interviewees [79, 80], it is crucial to draw the most accurate overview of the diversity of wild food items used by local people, especially in species-rich ecosystems in which plants, animals and human livelihoods are under increasing pressure.

## Conclusions

Our results show that choosing an appropriate method (or combination of methods) is essential for assessing wild food plant knowledge and consumption. Ethnobotanical inventories that leave room for informants to point out edible plants that do not appear on predefined lists are essential to capture the full diversity of edible forest species. Our *in-situ* inventories yielded a wide variety of WEP that were not covered by our *ex-situ* interview methods, revealed details on local patterns of foraging and challenged the general ideas of knowledge loss and underutilization of wild food plants in the region. Our mixed methods approach shows the importance of cross-referencing data, not only between different types of interviews, but also between interviews and direct observation during forest trips, for a better assessment of the diversity, consumption frequency, trade and conflicting uses of WEP. We therefore recommend that wild plant knowledge and use should be assessed through the use of a mixed methods approach. Using 'open' walk-in-the-woods surveys, in which informants are encouraged to mention any useful plant they know or randomly encounter, after which they are asked when they last used it, provides a wider variety of data. Employing the walk-in-the-woods technique merely to supply specimens for previously composed lists of useful plants from literature or interviews limits the capacity of this powerful technique to assess wild plant knowledge and use. Freelisting and dietary recalls can be used to provide quantitative data and a more general overview, but researchers should not limit themselves to such methods if they want to capture the full diversity of plants known and used, especially in the case when biased conclusions may have large implications for people's future livelihood, culture and wellbeing.

## Supporting information

**S1 Table. Wild edible plants reported during the walk-in-the-woods, free listing and dietary recalls in Le Bosquet and Kungu, southeast Cameroon.**
(XLSX)

**S1 File. Data collection protocol for *ex-situ* interviews and *in-situ* interviews.**
(DOCX)

## Acknowledgments

We would like to thank our assistants Appolinaire Ambassa, Ernest Isidore Simpoh and driver Alain Hyppolite Fezeu for their help with the fieldwork. We are grateful to Professor Bonaventure Sonké for his support in Cameroon. David Harris (Edinburgh herbarium); Jan Wieringa, Paul Maasand Carel Jongkind (Naturalis) and Marc Sosef (Meise Botanical Gardens) helped us to identify our specimens. Finally, our greatest gratitude goes to all Baka children, women and men with whom we have lived and worked. Thank you for your trust, hospitality and generous hearts.

## Author Contributions

**Conceptualization:** Sandrine Gallois, Thomas Heger, Amanda Georganna Henry, Tinde van Andel.

**Data curation:** Sandrine Gallois, Thomas Heger.

**Formal analysis:** Thomas Heger, Tinde van Andel.

**Funding acquisition:** Thomas Heger, Amanda Georganna Henry, Tinde van Andel.

**Investigation:** Sandrine Gallois, Amanda Georganna Henry.

**Methodology:** Sandrine Gallois, Thomas Heger, Tinde van Andel.

**Project administration:** Amanda Georganna Henry.

**Writing – original draft:** Sandrine Gallois, Thomas Heger, Tinde van Andel.

**Writing – review & editing:** Sandrine Gallois, Thomas Heger, Amanda Georganna Henry, Tinde van Andel.

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
