## [Editor Report · Decision Letter 0]

16 Dec 2020

PONE-D-20-37238

Importance of choosing appropriate methods for assessing wild food plant knowledge and use: a case study among the Baka in Cameroon.

PLOS ONE

Dear Dr. Sandrine Gallois

Thank you for submitting your manuscript to PLOS ONE. After careful consideration, we feel that it has merit but does not fully meet PLOS ONE’s publication criteria as it currently stands. Therefore, we invite you to submit a revised version of the manuscript that addresses the points raised during the review process.

We look forward to receiving your revised manuscript.

Kind regards,

Muhammad Ishtiaq

Academic Editor

PLOS ONE

Additional Editor Comments:

Paper needs followings changes/improvements and data provisions; in order to proceed it for further.

(1) needs to improve English language, there are many mistakes in the article.(letter of English experts reviewed as supporting needed.

(2) letter of Ethical and ISE and other institution permission letters are required.

(3) The references cited in text and reference section needs to be rechecked as it has many mistakes in it.

(4) Figure of study area needed.

(5) model of questionnaire needed.

(6) Families names are not correct, recheck and resend table 1.

(7) Provide ubiquitous herbarium numbers for all plants collected. it has two types of numbering.

(8) Figures are dim/blurred, send high density/good quality figures.
---

## [Author Response · Author response to Decision Letter 0]

6 Jan 2021

Anna Fodor Leiden, 6 January, 2021

Academic Editor

PLOS ONE

Dear Dr. Fodor,

Thank you for allowing us to adapt our manuscript PONE-D-20-37238, entitled “The importance of choosing appropriate methods for assessing wild food plant knowledge and use: a case study among the Baka in Cameroon”. We have followed your suggestions and hope that the manuscript now fits the submission standards of PLoS ONE. 

We have submitted a revised version of our manuscript labeled as “revised manuscript with track changes” and a clean copy (unmarked) labeled as “manuscript”, as well as the response to the Editor's comments, Dr Muhammad Ishtiaq, in the document "Response to Reviewer" with a point by point answers.

Below, we list each point made by the academic editor, and our respective answer. 

Additional Editor Comments:

Paper needs followings changes/improvements and data provisions; in order to proceed it for further.

(1) needs to improve English language, there are many mistakes in the article.(letter of English experts reviewed as supporting needed.

 Answer to comment #1: Our manuscript was checked by someone with ample experience in publishing scientific articles in the English language.

(2) Letter of Ethical and ISE and other institution permission letters are required.

 Answer to comment #2: The International Society for Ethnobiology does not issue permission letters and does not have an ethnical board: it just published standards for researchers, to which we adhered. However, our research was carried out within the ERC project “HARVEST: Plant Foods in Human Evolution; Factors affecting the harvest of nutrients from the floral environment”. For this research project, our co-author Amanda Henry received approval from the Ethical Board of the Faculty of Medicine of the University of Leipzig, Germany. We have added this letter to our submitted files and listed the HARVEST project in the funding. Moreover, before we started our fieldwork, we received permission from the National Committee of Ethics for the Research of Human Health (CNERSH) of the Cameroonian government. We also added this letter to the submitted files.

(3) The references cited in text and reference section needs to be rechecked as it has many mistakes in it.

Answer to comment #3: We have checked all references in the text and reference list and adapted them to the style of PLoS ONE. 

(4) Figure of study area needed.

Answer to comment #4: We included a map of the study area (Figure 1)

(5) Model of questionnaire needed.

Answer to comment #5: We included the model of our questionnaire in our Supporting Information (Supplementary file 1).

(6) Families names are not correct, recheck and resend table 1.

Answer to comment #6: Family and author names of the species are now listed in the Supporting Information (Supplementary Table 1). All scientific names were checked and updated suing Plants of the World Online.

(7) Provide ubiquitous herbarium numbers for all plants collected. it has two types of numbering.

Answer to comment #7: Herbarium numbers for all plants collected are listed in Supplementary Table 1, but there are two acronyms (WTH and SG), referring to the two main collectors: Willem Thomas Heger and Sandrine Gallois. We had to follow the Naturalis guidelines of specimen collection for this study.

(8) Figures are dim/blurred, send high density/good quality figures.

Answer to comment #8: We have now uploaded high quality figures following the guidelines of PLoS ONE.

On behalf of our co-authors, we hope that our manuscript is now suitable to be considered for review,

Yours sincerely, 

Sandrine Gallois and Tinde van Andel

---

## [Decision Letter · Decision Letter 1]

2 Feb 2021

The importance of choosing appropriate methods for assessing wild food plant knowledge and use: a case study among the Baka in Cameroon.

PONE-D-20-37238R1

Dear Dr. Sandrine Gallois

We’re pleased to inform you that your manuscript has been judged scientifically suitable for publication and will be formally accepted for publication once it meets all outstanding technical requirements.

Kind regards,

Dr. Muhammad Ishtiaq Ch.

Academic Editor

PLOS ONE

Additional Editor Comments (optional):

Reviewers' comments:

Reviewer's Responses to Questions

**Comments to the Author**

1. If the authors have adequately addressed your comments raised in a previous round of review and you feel that this manuscript is now acceptable for publication, you may indicate that here to bypass the “Comments to the Author” section, enter your conflict of interest statement in the “Confidential to Editor” section, and submit your "Accept" recommendation.

Reviewer #1: All comments have been addressed

Reviewer #2: All comments have been addressed

Reviewer #3: All comments have been addressed

Reviewer #4: All comments have been addressed

Reviewer #5: All comments have been addressed

Reviewer #6: All comments have been addressed

Reviewer #7: All comments have been addressed

2. Is the manuscript technically sound, and do the data support the conclusions?

Reviewer #1: Yes

Reviewer #2: Yes

Reviewer #3: Yes

Reviewer #4: Yes

Reviewer #5: Yes

Reviewer #6: Yes

Reviewer #7: Yes

3. Has the statistical analysis been performed appropriately and rigorously? 

Reviewer #1: Yes

Reviewer #2: Yes

Reviewer #3: N/A

Reviewer #4: I Don't Know

Reviewer #5: Yes

Reviewer #6: Yes

Reviewer #7: Yes

4. Have the authors made all data underlying the findings in their manuscript fully available?

Reviewer #1: Yes

Reviewer #2: Yes

Reviewer #3: Yes

Reviewer #4: Yes

Reviewer #5: Yes

Reviewer #6: Yes

Reviewer #7: Yes

5. Is the manuscript presented in an intelligible fashion and written in standard English?

Reviewer #1: Yes

Reviewer #2: Yes

Reviewer #3: Yes

Reviewer #4: Yes

Reviewer #5: Yes

Reviewer #6: Yes

Reviewer #7: Yes

6. Review Comments to the Author

Reviewer #1: I have reviewed the article and have found it sound and valid for publication. The revisions have addressed the comments made by the reviewers.

Reviewer #2: Previous Suggestions and comments are incorporated, however, article format need to be improved according to PLOS format.

Reviewer #3: (No Response)

Reviewer #4: I recommend acceptance of the paper for publication because authors have revised the paper successfully.

Reviewer #5: PONE-D-20-37238R1

The importance of choosing appropriate methods for assessing wild food plant knowledge and use: a case study among the Baka in Cameroon.

Reviewer Response:

This manuscript needs following changes/improvements in order to proceed further:

1. In the abstract section, the line number 37 should be rephrased as; We discussed the limitation and strength of these different methods for investigating the diversity of wild food plant; their available knowledge and usage.

2. Grammatical errors should be improved e.g. use of helping verbs, placements of punctuation signs.

a. Line number 110: Plant knowledge of a certain group of people; should be corrected as; plant knowledge by a certain group of people.

b. Line number 112: The specimen of species mentioned during free listing should be corrected as; specimen of species identified during free listing.

c. Line number 115: All useful plants they know and/or use should be corrected as; all useful plants they know or their usage.

3. The line number 38 should be made understandable as; Hence, this analysis will help to further investigate more information regarding wild food plant and their usage frequency, through mixed approach the combines in-situ and ex-situ methods.

4. The line number 70; According to Ngome et al.; as reference number is already given, so rearrange the sentence like; According to another study reported,… .

5. From line number 100 to 123; the list of different ethnobotanical method used in this investigation should be included in the methodology section.

6. Line number 156: the sentence should be like; how do different methods for assessing WEP used in this study vary in their results?

7. It is suggested to have a scheme of study i.e. comprehensive framework under which the study is designed and executed e.g. a flow chart or a step by step methodology (figure) followed in this investigation.

8. Line number 180: our inventory of wild food plant knowledge, use and commercialization should be corrected as; our inventory of wild food plant knowledge, their usage and its commercialization.

9. All the tables in the manuscript should be justified as, “Centered- alignment” for all cells of the table; or keep the same alignment for all the tables.

10. Table 1; number of participants should be corrected as; (number of participants= n).

11. Line number 274; kindly mention the figure number.

12. Line number 354; it is recommended to rephrase the heading as; WEP parts used and their preparation method.

13. Line number 336; as this line is a part of table number 2, so try to align it on the same page with the table.

14. Table number 3; please elaborate what denotes “y” here.

15. Line number 525- 541; if this information can be presented in tabular or graphical form/as there is no reference given for any table or figure, that where they are presented.

16. Line number 572; on a population level in our sample should be like; on a population level in our sample collection.

17. Line number 575; kindly correct the formatting mistake.

18. It is suggested that “The accuracy of different methods used in this investigation can be presented either in a graphical or tabular form to highlight their efficiency”.

Reviewer #6: It is suggested to list plant species which are used as folk medicine in a separate list and those which are consumed as dietary by people. Are any plants that have not been reported by plant taxonomists?

Reviewer #7: In this study, Gallois et al present a case study suggesting the importance of appropriate methods for assessing wild food plant knowledge and consumption.

The comments raised by the previous reviewers were very pertinent, and authors have satisfactorily addressed all but one comment, i.e. the quality of figures is still poor and needs to be improved. Authors must provide high quality pictures. Other than that, the manuscript is in good shape and acceptable for publication.

7. PLOS authors have the option to publish the peer review history of their article (what does this mean?). If published, this will include your full peer review and any attached files.

Reviewer #1: **Yes: **Muhammad Azeem Abbas

Reviewer #2: No

Reviewer #3: **Yes: **Dr. SOHAIL

Reviewer #4: No

Reviewer #5: No

Reviewer #6: **Yes: **Kadhim M. Ibrahim

Reviewer #7: No

---

## [Editor Report · Acceptance letter]

8 Feb 2021

PONE-D-20-37238R1 

The importance of choosing appropriate methods for assessing wild food plant knowledge and use: a case study among the Baka in Cameroon. 

Dear Dr. Gallois:

I'm pleased to inform you that your manuscript has been deemed suitable for publication in PLOS ONE. Congratulations! Your manuscript is now with our production department. 

Kind regards, 

on behalf of

Dr. Muhammad Ishtiaq 

Academic Editor

PLOS ONE